# The role of sex and handedness in the performance of the smartphone-based Finger-Tapping Test

**Felipe André Costa Brito[1], Luis Carlos Pereira Monteiro [1], Enzo Gabriel Rocha Santos[2], Ramon Costa de Lima[1], Bruno Lopes Santos-Lobato[3], André Santos Cabral[4], Bianca Callegari [5], Anselmo de Athayde Costa e Silva[6], Givago Silva Souza [1,7] ***

**1** Instituto de Ciências Biológicas, Universidade Federal do Pará, Belém, Brazil, **2** Instituto de Ciências Exatas, Universidade Federal do Pará, Belém, Brazil, **3** Instituto de Ciências da Saúde, Universidade Federal do Pará, Belém, Brazil, **4** Centro de Ciências Biológicas e da Saúde, Universidade do Estado do Pará, Belém, Brazil, **5** Laboratório de Estudos do Movimento Humano, Instituto de Ciências da Saúde, Universidade Federal do Pará, Belém, Brazil, **6** Programa de Pós-graduação em Ciências do Movimento Humano, Instituto de Ciências da Saúde, Universidade Federal do Pará, Belém, Brazil, **7** Núcleo de Medicina Tropical, Universidade Federal do Pará, Belém, Brazil

* givagosouza@ufpa.br

**Data Availability Statement:** All the data of this study are available at https://figshare.com/articles/dataset/FingerTapping_results_rar/21498279.

## Abstract

The Finger Tapping Test (FTT) is a classical neuropsychological test that assesses motor functioning, and recently it has been employed using smartphones. For classical protocols, it has been observed that sex and handedness influence the performance during the test. By assessing the influence of sex and handedness on the test, it is possible to adjust the performance measurements to ensure the validity of test results and avoid sex- and handedness-related bias. The present study aimed to evaluate the influence of sex and handedness on smartphone-based FTT performance. We developed an Android application for the FTT and recruited 40 males and 40 females to carry out three spatial designs on it (protocols I, II, and III). Participants' performance was measured using the global, temporal, and spatial parameters of the FTT. We observed that for the performance in protocol I, handedness had a significant influence on global and temporal variables, while the interaction between handedness and sex had a greater influence on spatial variables. For protocols II and III, we observed that handedness had a significant influence on global, temporal, and spatial variables compared to the other factors. We concluded that the smartphone-based test is partly influenced by handedness and sex, and in clinical implications, these factors should be considered during the evaluation of the smartphone-based FTT.

## Author summary

The finger tapping test is a test that measures how well someone can tap a surface quickly. It has been used for many years to evaluate motor function in people with neurological diseases. In the past, telegraph keys, computer mice, and keyboards were used to do the test. However, nowadays smartphones with touchscreens are often used. Before this

**Funding:** This work was supported by research grants from the Brazilian funding agencies: CNPq Edital Universal (#431748/2016-0, GSS), and Programa de Apoio à Publicação Qualificada from Federal University of Pará (GSS). GSS and BLSL are CNPq Fellow and receive productivity grants (protocol #408288/2022-1). The funders had no role in study design, data collection and analysis, decision to publish, or preparation of the manuscript.

**Competing interests:** The authors have declared that no competing interests exist.

technology can be widely used in clinics, it's important to investigate factors that could affect the test results. Two factors that might impact the results are a person's sex and handedness, which is their preference for using one hand over the other. In this study, we investigated whether sex and handedness influence how well someone performs in the finger tapping test using a smartphone. We found that both sex and handedness play a role in how well someone performs, and in general, we observed that males using their dominant hand tend to perform the best on the test. These results suggest that it's important to consider a person's sex and handedness when interpreting the results of the smartphone-based finger tapping test.

## Introduction

The Finger-Tapping Test (FTT) is a widely used neuropsychological test that assesses motor functioning, lateralized coordination, and motor speed [1]. This test involves the subject tapping their index finger on a surface as quickly as possible for a set period while keeping their hand resting on a board [2]. The original version of the test required participants to complete 10 trials with each hand, which made the test time-consuming. The original version measured the number of taps made in a set amount of time, but modifications to the test protocol, instruments, and analysis have been suggested over the years to shorten the test duration [3], adapt it to new technologies [2], and extract more features [4] from motor performance during the test.

The FTT is a widely used tool in both clinical and research settings for assessing various conditions that affect motor function and neurological health. It has been utilized to evaluate the effects of traumatic brain injury, stroke, Parkinson's disease, multiple sclerosis, and other neurological disorders [2,5–7]. This simple, non-invasive, and cost-effective test can be administered in various settings and provides valuable insights into the underlying neurological mechanisms that contribute to motor performance.

Initially, the telegraph key was commonly used to record finger taps during the FTT, however, alternative technologies have been proposed, such as the computer mouse [2], keyboard [8], and image-based motion [6]. In recent years, touchscreens from smartphones have emerged as a novel surface for performing the FTT. This approach offers several advantages and potential benefits over traditional methods. The widespread availability and portability of smartphones make it easier to administer the test remotely or in non-clinical settings. The use of smartphones allows for standardization of test administration, as the device can be programmed to deliver the test in a consistent and reproducible manner. Furthermore, smartphones enable precise measurement of tapping speed and duration, as well as facilitating data collection and analysis through digital recording and storage of test results. In summary, the use of touchscreens from smartphones provides a promising and convenient approach to administering the FTT, with potential implications for improving assessment and monitoring of neurological conditions.

Typically, the FTT quantifies performance by measuring the total number of taps and temporal variables such as frequency of taps, interval between taps, and duration of taps [2]. However, the emergence of smartphone touchscreen technology has opened up the possibility of recording the position of the screen where finger taps were performed and extracting spatial features of the task, such as the distance between tap coordinates and total distance of finger movements. Lee et al. [4] pioneered the evaluation of spatial parameters obtained from a smartphone-based FTT to assess bradykinesia in patients with Parkinson's disease compared

to healthy individuals. Subsequent studies using smartphone-based FTT have also applied the test in clinical conditions, mostly in patients with Parkinson's disease, comparing global, temporal, and spatial variables between controls and patients [9–16]. By incorporating spatial parameters into FTT assessment, smartphone technology provides a promising avenue for more comprehensive and accurate evaluation of motor function in clinical settings.

Given the advantages that smartphones offer as a tool for performing FTT, it is crucial to examine potential factors that may interfere with motor performance during the test. Sex and handedness, i.e., preference or ability to use one hand rather than the other across a range of common activities, are two such factors that have been shown to affect performance [17,18]. By assessing the influence of sex and handedness on finger tapping, researchers can determine whether adjustments are necessary to ensure the accuracy and validity of test results. This approach can help to mitigate bias and ensure that the FTT is applicable to individuals of all sexes and handedness. Ultimately, such evaluations can enhance the reliability and clinical utility of FTT for assessing motor function.

To date, no study has examined how participant characteristics such as sex and handedness may influence performance on smartphone-based FTT. In the current study, we sought to investigate the influence of sex and handedness on global, temporal, and spatial parameters obtained from smartphone-based FTT. By shedding light on these issues, our findings could inform the development of more accurate and reliable assessments of motor function in clinical and research settings.

## Methods

### Ethical considerations

The present study was approved by the Research and Ethics Committee of the Federal University of Pará (report # 6.036.494). All participants were informed about the experimental procedures and signed an informed consent form to participate in the study. All procedures are in accordance with the Declaration of Helsinki complying with all relevant ethical regulations.

### Participants

Eighty right-handed participants (40 males and 40 females) comprised the sample of the present study, ranging between 18 and 50 years (males: 31.7 ± 8.04 years; females: 34.63 ± 9.37 years). The participants in this study were recruited through convenience sampling from a university population. The handedness of the participants was evaluated using the short version of the self-report Edinburgh handedness inventory [19]. Only right-handed participants were included, because there are differences in the cortical organization and motor performance between right-handed and left-handed people [20,21]. All participants had no history of degenerative diseases or use of drugs that could compromise movements.

### Protocols of a smartphone-based finger-tapping test

An Android OS application, *Momentum Touch app*, was programmed using Android Studio and installed on a Samsung Galaxy S10 Plus smartphone. The smartphone had a 6.4-inches Dynamic AMOLED, QHD+ Display (3040 x 1440 Pixels), HDR10 19:9 Aspect Ratio, Gorilla Glass 6, 536 pixels per inch, 120 Hz resolution to touch capture.

For the test, the smartphone was placed on a table in landscape orientation related to the participant, with the top of the smartphone facing left. Participants performed the test while sitting in a chair with the foot touching the ground, and a chair height that allowed the hand to rest on the table with the elbow in 90˚ flexion, pronated forearm and the wrist at 0˚ flexion.

Prior to the experiments, all participants underwent a training test and received instructions by the same experimenter to ensure they were familiar with the procedures and tasks. This step was taken to minimize any potential confusion or misunderstandings during the actual experiment and to ensure that all participants had a consistent level of understanding and preparation. All participants carried out three modified protocols of FTT based on smartphones:

*Protocol I*: it was designed to evaluate the performance of the participant targeting a single-centered area of the smartphone. Protocol I is the closer condition to the classical FTT, in which the participant was asked to tap repeatedly on the same target during the test. For Protocol I, the participant's task was to tap the index finger in the center area as fast as possible. Taps outside the central target were considered errors in the task.

*Protocol II*: it was designed to evaluate the performance of the participant in a task of alternating taps on both sides of the smartphone without a target on the screen. For Protocol II, the participant's task was to tap alternately as fast as possible on both sides of the central division (region 1 on the left and region 2 on the right). One consecutive tap on the same side of a previous tap was considered an error in the task.

*Protocol III*: it was designed to evaluate the performance of the participant in a task of alternating taps in targets centered on both sides of the smartphone. For protocol III, the participant's task was to tap alternatively as fast as possible in two central areas on both sides of the central division (region 1 on the left and region 2 on the right). One consecutive tap on the same side of a previous tap or a tap out of the central areas was considered an error in the task.

Fig 1 shows the spatial properties of the visual representation of the smartphone screen for each FTT protocol.

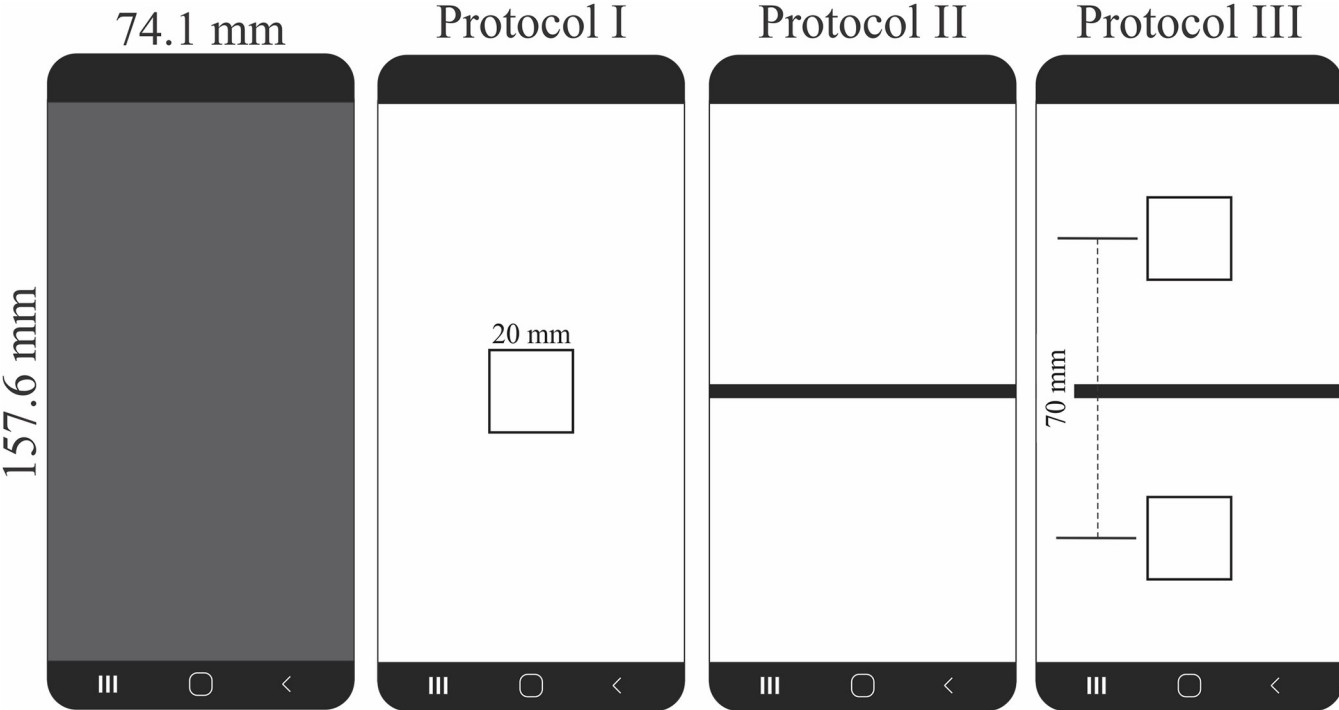

**Fig 1. Smartphone dimensions of the model used in the present study and design of the three protocols proposed.** Protocol I had a centered target (20 mm), Protocol II had a midline separating two equal areas without a restricted target, and Protocol III had a midline separating two equal sides with centered targets on each side 70 mm apart.

The test duration for each protocol was 30 seconds, with 1 minute interval between trials. The hand and the protocol to be tested were randomly chosen. Total duration of all experiments was 9.5 minutes. Our choice for an abbreviated version of FTT is to reduce the total testing duration similarly other researchers have tried [3,22].

### Data analysis

The smartphone recorded the spatial coordinates of the screen where there was a tap, the moment of the tap, and a Boolean variable indicating whether the tap was done in the target area and exported a text file to be analyzed in routines in the MATLAB/OCTAVE or R environment. The following parameters were used to quantify the performance of the participant in each FTT protocol previously described:

i. *Global parameters*:

- Total number of taps.

- Number of errors.

ii. *Temporal parameters*:

- Median frequency of taps.

- Maximum frequency of taps.

- Minimum frequency of taps.

iii. *Spatial parameters*:

- Total displacement: the sum of all distances between the taps. The distance between the taps was calculated following Eq 1.

$$Distance = \sqrt{\left(x_i - x_{i+1}\right)^2 + \left(y_i - y_{i+1}\right)^2} \qquad (1)$$

Where $x_i$ and $y_i$ are the screen coordinates of the *ith* tap.
- Area of a confidence ellipse covering 95% of the balance sway (Eq 2) that was fitted using the MATLAB/OCTAVE code.

$$area = \pi \times D \times d \qquad (2)$$

Where $D$ is the major axes of the ellipse, and $d$ is the minor axis of the ellipse.
- Major axes of the ellipse obtained from the ellipse model that best fit the body sway data.
- Minor axes of the ellipse obtained from the ellipse model that best fitted to the body sway data.
- Kernel density estimation to calculate the peak density and bandwidths for the X- and Y-axis performed using kde2d() in the R package MASS.

For the protocols II and III, the ellipse area, major and minor axis were estimated for the tap coordinates from both screen sides.

### Statistics

To test the influence of sex and handedness on the FTT performance, we applied Linear Mixed Models to the parameters extracted from the FTT dataset. Subject ID was included as a random effect. Type-III ANOVA was used to estimate the *F*-statistic and *p*-value for each variable in the models. To verify model assumptions, we applied visual inspection followed by a

Shapiro-Wilk normality test and a Bartlett test of homogeneity of variance on models' residuals. For these models, we calculated the partial $\eta^2$ to compute the effect size. In the cases the model did not fit well, we applied a Friedman repeated measures test followed by a Dunn multiple comparison test. All analyzes were performed using R and the following packages: lme4, lmerTest, and performance. A confidence level of 0.05 was considered for all statistical tests.

## Results

### Results of protocol I

Fig 2A–2D shows the mean temporal series (± confidence interval) of the tapping frequency during the test execution for male and female participants using dominant and nondominant hands. The mean frequency was maximal at the beginning of the test and decreased along the test duration. Additionally, Fig 2E shows an example of the spatial distribution of the taps for a representative participant.

Table 1 shows the results of protocol I we proposed for FTT. Number of errors was not significantly influenced by sex and handedness and no significant interaction between the factors was observed (Friedman $Q_{(3,40)} = 1.2$, $p = 0.75$). Sex significantly influenced the total number of taps (ANOVA $F_{(1,78)} = 17.97$, $p < 0.001$), median frequency (ANOVA $F_{(1,78)} = 17.29$,

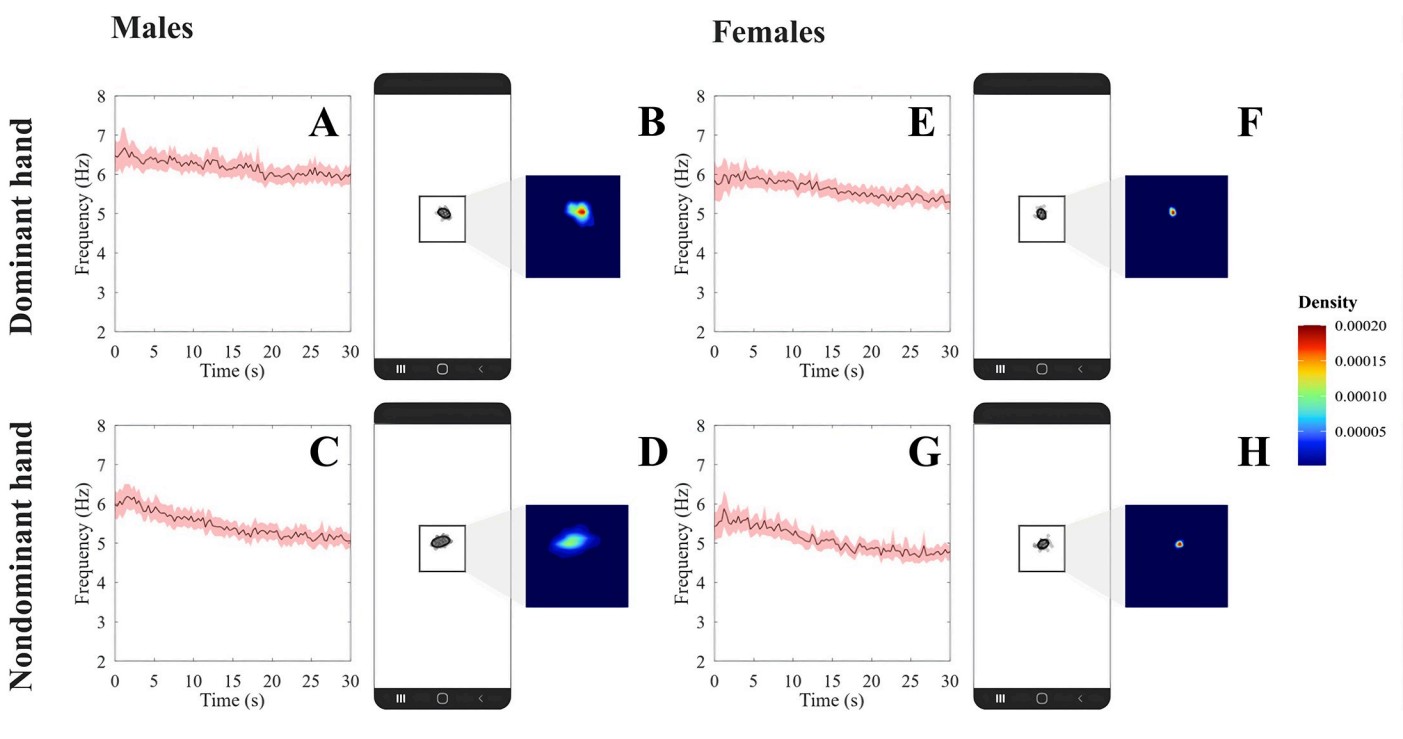

**Fig 2. Performance in Protocol I of the Finger-Tapping Test proposed in the present study.** (A) is the FTT frequency as a function of time for dominant hand from males; (B) Spatial kernel density estimation plot generated by the FTT executed from dominant hand from males; (C) is the FTT frequency as a function of time for non-dominant hand from males; (D) Spatial kernel density estimation plot generated by the FTT executed from non-dominant hand from males. (E) is the FTT frequency as a function of time for dominant hand from females; (F) Spatial kernel density estimation plot generated by the FTT executed from dominant hand from females; (G) is the FTT frequency as a function of time for non-dominant hand from females; (H) Spatial kernel density estimation plot generated by the FTT executed from non-dominant hand from females. Black crosses represent valid taps, red crosses represent non-valid taps and red ellipsis are the 95% confidence ellipsis fitted to the data.

**Table 1. Descriptive statistics [median (interquartile range)] of the global, temporal, and spatial parameters for Protocol I.**

| | Female | | Male | | |
| --- | --- | --- | --- | --- | --- |
| | **DH** | **NDH** | **DH** | **NDH** | **Significant effect size** |
| *Global parameters* | | | | | |
| Total number of taps | 169 (22.5) | 148.5 (21.5) | 184 (17.5) | 162.5 (23) | Sex*, handedness* |
| Number of errors | 0 (0) | 0 (0) | 0 (0) | 0 (0) | None |
| *Temporal Parameters* | | | | | |
| Median frequency | 5.75 (0.73) | 5.06 (0.60) | 6.23 (0.49) | 5.46 (0.77) | Sex*, handedness*, Interaction*** |
| Maximum frequency | 7.72 (1.83) | 7.12 (4.18) | 8.55 (2.43) | 8.00 (1.64) | None |
| Minimum frequency | 3.62 (1.05) | 4.57 (0.79) | 4.48 (1.33) | 4.10 (0.86) | Handedness**, Interaction** |
| *Spatial parameters* | | | | | |
| Total displacement | 7529.70 (1081.28) | 6682.23 (846.56) | 8171.38 (828.49) | 7243.70 (1057.14) | Sex*, handedness* |
| Major axis | 2.70 (1.28) | 3.09 (1.07) | 2.98 (1.08) | 2.84 (0.90) | Interaction* |
| Minor axis | 1.74 (0.65) | 2.16 (0.70) | 1.99 (0.61) | 1.97 (0.79) | Interaction* |
| Area | 14.03 (9.74) | 22.25 (13.21) | 19.24 (13.24) | 18.55 (9.46) | Interaction* |
| Peak density | 3.03 (2.17) | 2.09 (1.78) | 2.52 (2.3) | 1.99 (1.46) | None |
| Bandwidth X | 38.52 (15.90) | 45.42 (20.60) | 39.18 (19.65) | 42.56 (18.16) | Handedness* |
| Bandwidth Y | 33.38 (14.00) | 40.26 (16.08) | 35.82 (18.91) | 41.96 (22.94) | Handedness* |

DH: dominant hand; NDH: nondominant hand.

*p < 0.001

**0.001 ≤ p < 0.01

***0.01 ≤ p < 0.05

$p < 0.001$), total displacement (ANOVA $F_{(1,78)} = 16.42$, $p < 0.001$). Handedness significantly influenced the total number of taps (ANOVA $F_{(1,78)} = 119.63$, $p < 0.001$), median frequency (ANOVA $F_{(1,78)} = 149.55$, $p < 0.001$), minimum frequency (ANOVA $F_{(1,78)} = 7.47$, $p = 0.007$), total displacement (ANOVA $F_{(1,78)} = 116.31$, $p < 0.001$), bandwidth X (ANOVA $F_{(1,78)} = 18.22$, $p < 0.001$) and bandwidth Y (ANOVA $F_{(1,78)} = 30.16$, $p < 0.001$). The interaction between both factors was significant in the median frequency (ANOVA $F_{(1,78)} = 4.20$, $p = 0.04$), minimum frequency (ANOVA $F_{(1,78)} = 9.28$, $p = 0.003$), major axis (ANOVA $F_{(1,78)} = 88.28$, $p < 0.001$), minor axis (ANOVA $F_{(1,78)} = 96.67$, $p < 0.001$), and area of the ellipse (ANOVA $F_{(1,78)} = 141.09$, $p < 0.001$). In peak density variable was found significant differences (Friedman $Q_{(3,40)} = 18.8$, $p < 0.001$), in which for both sexes there was significant differences between dominant and non-dominant hands.

Global and temporal variables had more frequent significant effects and larger effect size on handedness, while spatial variables had larger and more frequent significant effect sizes of the interaction between the factors.

## Results of the protocol II

Fig 3A–3D shows the mean temporal series (± confidence interval) of the tapping frequency during the test execution for male and female participants using dominant and nondominant hands. Here, the mean frequency also reached higher values in the first 5 seconds of the test and decayed throughout the duration of the test. Fig 3E shows an example of the spatial distribution of the taps on both sides of the screen for a representative participant.

Table 2 shows the results of protocol II we proposed for FTT. Number of errors was not significantly influenced by sex and handedness and no significant interaction between the factors was observed (Friedman $Q_{(3,40)} = 3.06$, $p = 0.38$). Sex significantly influenced the total

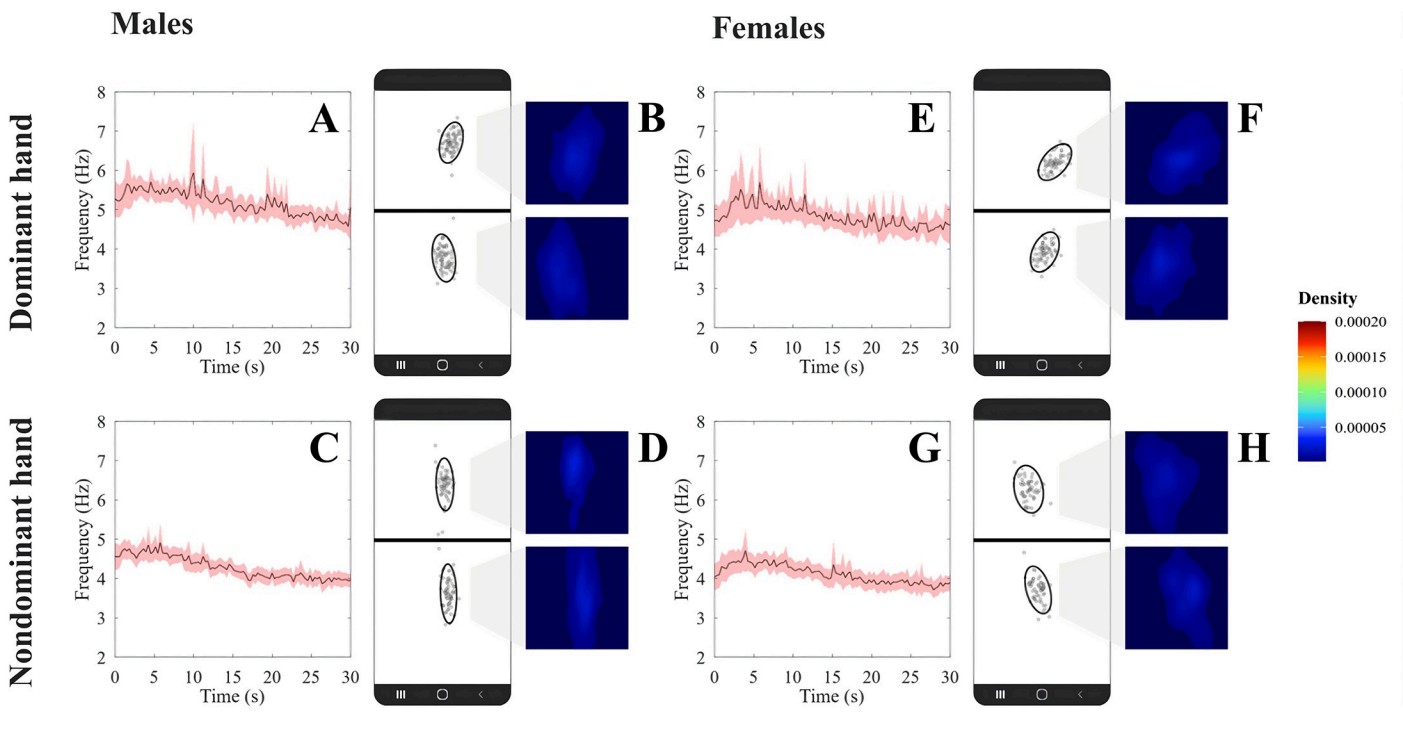

**Fig 3. Performance in Protocol II of the Finger-Tapping Test proposed in the present study.** (A) is the FTT frequency as a function of time for dominant hand from males; (B) Spatial kernel density estimation plot generated by the FTT executed from dominant hand from males; (C) is the FTT frequency as a function of time for non-dominant hand from males; (D) Spatial kernel density estimation plot generated by the FTT executed from non-dominant hand from males. (E) is the FTT frequency as a function of time for dominant hand from females; (F) Spatial kernel density estimation plot generated by the FTT executed from dominant hand from females; (G) is the FTT frequency as a function of time for non-dominant hand from females; (H) Spatial kernel density estimation plot generated by the FTT executed from non-dominant hand from females. Black crosses represent valid taps, red crosses represent non-valid taps and red ellipsis are the 95% confidence ellipsis fitted to the data.

displacement (ANOVA $F_{(1,78)}$ = 4.05, $p$ = 0.04), major axis 1 (ANOVA $F_{(1,78)}$ = 14.37, $p < 0.001$) and major axis 2 (ANOVA $F_{(1,78)}$ = 7.12, $p$ = 0.008). Handedness significantly influenced the total number of taps (ANOVA $F_{(1,78)}$ = 202.30, $p < 0.001$), median frequency (ANOVA $F_{(1,78)}$ = 393.71, $p < 0.001$), maximum frequency (Friedman $Q_{(3,40)}$ = 37.97, $p < 0.001$), minimum frequency (ANOVA $F_{(1,78)}$ = 6.85, $p$ = 0.01), total displacement (ANOVA $F_{(1,78)}$ = 214.67, $p < 0.001$), minor axis 1 (ANOVA $F_{(1,78)}$ = 6.57, $p$ = 0.01), area of the ellipse 1 (ANOVA $F_{(1,78)}$ = 5.39, $p$ = 0.02), bandwidth Y 1, major axis 2 (ANOVA $F_{(1,78)}$ = 12.64, $p < 0.001$), bandwidth Y 1 (ANOVA $F_{(1,78)}$ = 20.92, $p < 0.001$) and Y 2 (ANOVA $F_{(1,78)}$ = 21.80, $p < 0.001$). The interaction between both factors was significant in the median frequency (ANOVA $F_{(1,78)}$ = 5.52, $p$ = 0.02), minor axis 1 (ANOVA $F_{(1,78)}$ = 10.59, $p$ = 0.002) and minor axis 2 (ANOVA $F_{(1,78)}$ = 34.02, $p < 0.001$), area of the ellipse 1 (ANOVA $F_{(1,78)}$ = 11.66, $p$ = 0.001), and bandwidth Y 2 (ANOVA $F_{(1,78)}$ = 4.47, $p$ = 0.04).

Handedness showed more significant and larger effect sizes on global, temporal, and spatial variables than the other factors.

## Results of the protocol III

Fig 4A–4D shows the mean temporal series (± confidence interval) of the tapping frequency during the test execution for male and female participants using dominant and nondominant

**Table 2. Descriptive statistics [median (interquartile range)] of the global, temporal, and spatial parameters for Protocol II.**

| | Female | | Male | | |
| --- | --- | --- | --- | --- | --- |
| | DH | NDH | DH | NDH | Significant effect size |
| *Global parameters* | | | | | |
| Total number of taps | 138 (23.25) | 118 (21.5) | 153 (26) | 128.5 (17.5) | Handedness* |
| Number of errors | 0 (1.25) | 0 (1) | 1 (3.25) | 0 (2) | None |
| *Temporal Parameters* | | | | | |
| Median frequency | 4.80 (0.63) | 4.1 (0.85) | 5.22 (0.87) | 4.42 (0.57) | Handedness*, Interaction*** |
| Maximum frequency | 6.16 (0.84) | 5.80 (1.76) | 7.31 (8.82) | 5.71 (1.25) | None |
| Minimum frequency | 2.42 (2.12) | 2.32 (1.19) | 2.76 (1.95) | 2.37 (1.04) | Handedness*** |
| *Spatial parameters* | | | | | |
| Total displacement | 6289.24 (1271.43) | 5350.08 (964.54) | 6856.21 (794.73) | 5806.12 (807.19) | Sex***, Handedness* |
| Major axis 1 | 9.13 (4.79) | 9.62 (4.55) | 9.82 (5.03) | 10.88 (3.94) | Sex* |
| Minor axis 1 | 4.23 (1.88) | 4.36 (1.84) | 4.84 (3.46) | 4.68 (1.7) | Handedness***, Interaction** |
| Area 1 | 118.48 (71.68) | 118.48 (98.85) | 159.92 (162.77) | 174.25 (106.93) | Handedness***, Interaction** |
| Major axis 2 | 8.93 (3.78) | 7.86 (5.04) | 10.07 (4.46) | 10.96 (5.23) | Sex**, Handedness* |
| Minor axis 2 | 4.06 (1.83) | 4.35 (2.33) | 4.69 (1.98) | 4.41 (1.84) | Interaction* |
| Area 2 | 102.96 (102.83) | 125.17 (83.21) | 131.22 (89.97) | 169.00 (109.95) | None |
| Peak density area 1 | 2.95 (1.51) | 2.22 (0.90) | 2.41 (1.69) | 2.10 (1.3) | None |
| Bandwidth X Area 1 | 97.94 (38.35) | 106.06 (38.89) | 103.73 (41.87) | 109.23 (48.45) | None |
| Bandwidth Y area 1 | 162.27 (53.23) | 192.02 (44.45) | 184.88 (61.71) | 207.70 (82.41) | Handedness*, Interaction*** |
| Peak density area 2 | 2.7 (2.1) | 2.4 (1.7) | 2.6 (2.2) | 2.2 (1.5) | None |
| Bandwidth X area 2 | 96.59 (44.01) | 109.45 (44.87) | 106.14 (42.83) | 107.31 (37.55) | None |
| Bandwidth Y area 2 | 163.03 (74.23) | 184.21 (61.30) | 161.28 (54.73) | 206.13 (110.91) | None |

DH: dominant hand; NDH: nondominant hand.

*p < 0.001

**0.001 ≤ p < 0.01

***0.01 ≤ p < 0.05

hands. The tapping frequency was kept constant during the entire test in this protocol. Fig 4E shows an example of the spatial distribution of the taps in both targets for a representative participant.

Table 3 shows the results of protocol III we proposed for FTT. Number of errors was not significantly influenced by sex and handedness and no significant interaction between the factors was observed (Friedman $Q_{(3,40)} = 6.81$, $p = 0.08$). Sex significantly influenced the minor axis 1 (ANOVA $F_{(1,78)} = 11.87$, $p < 0.001$) and minor axis 2 (ANOVA $F_{(1,78)} = 7.63$, $p = 0.007$), area of the ellipse 1 (ANOVA $F_{(1,78)} = 5.16$, $p < 0.001$) and 2 (ANOVA $F_{(1,78)} = 9.70$, $p = 0.002$). Handedness significantly influenced the total number of taps (ANOVA $F_{(1,78)} = 118.71$, $p < 0.001$), median frequency (ANOVA $F_{(1,78)} = 290.15$, $p = 0.03$), minimum frequency (ANOVA $F_{(1,78)} = 26.80$, $p < 0.001$), total displacement (ANOVA $F_{(1,78)} = 112.87$, $p < 0.001$), major axis 1 (ANOVA $F_{(1,78)} = 4.66$, $p = 0.03$), bandwidth X 1 (ANOVA $F_{(1,78)} = 25.73$, $p < 0.001$), bandwidth Y 1 (ANOVA $F_{(1,78)} = 17.56$, $p < 0.001$), minor axis 2 (ANOVA $F_{(1,78)} = 18.35$, $p < 0.001$), area of the ellipse 2 (ANOVA $F_{(1,78)} = 4.84$, $p = 0.03$), bandwidth X 2 (ANOVA $F_{(1,78)} = 17.27$, $p < 0.001$), and bandwidth Y 2 (ANOVA $F_{(1,78)} = 15.77$, $p < 0.001$). The interaction between both factors was significant in the median frequency (ANOVA $F_{(1,78)} = 7.61$, $p = 0.007$) and major axis 1 (ANOVA $F_{(1,78)} = 6.02$, $p = 0.02$).

Handedness showed more significant and larger effect sizes on global, temporal, and spatial variables than the other factors.

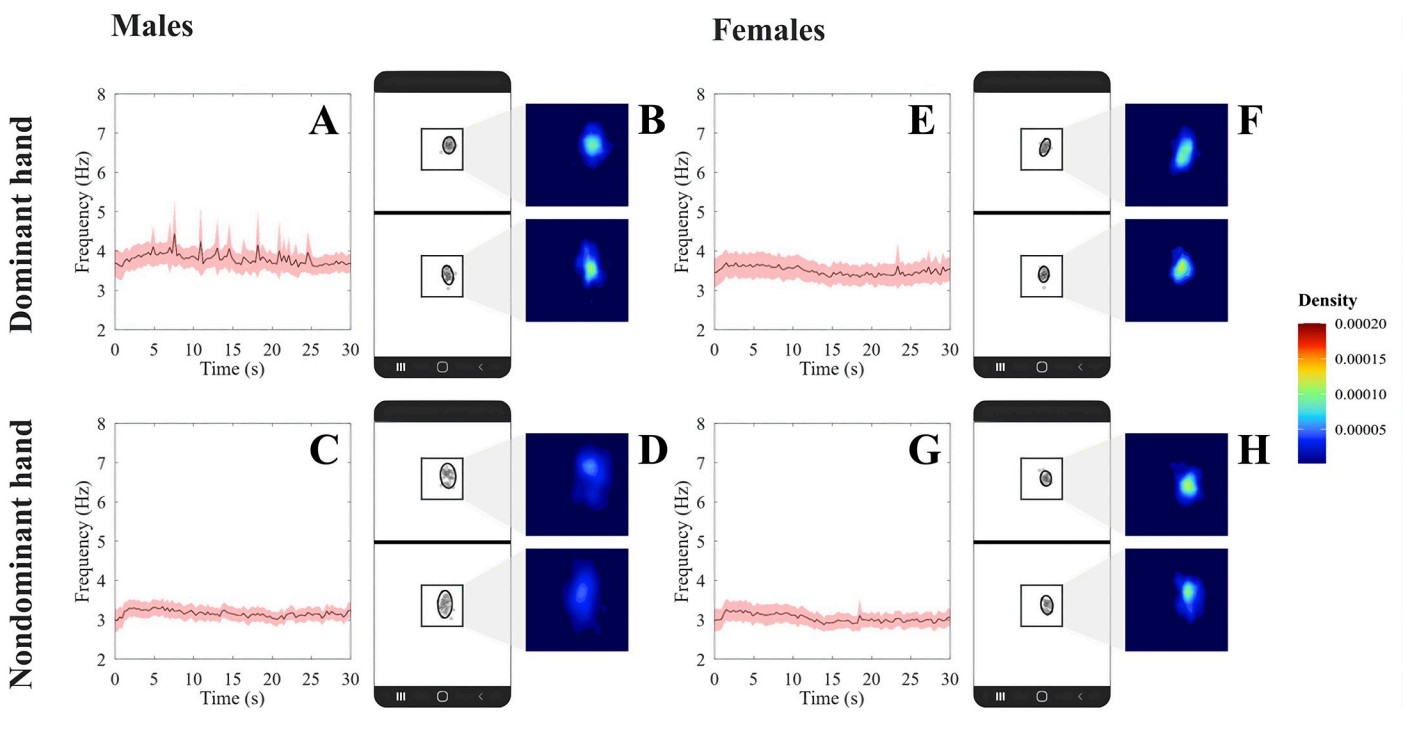

**Fig 4. Performance in Protocol III of the Finger-Tapping Test proposed in the present study.** (A) is the FTT frequency as a function of time for dominant hand from males; (B) Spatial kernel density estimation plot generated by the FTT executed from dominant hand from males; (C) is the FTT frequency as a function of time for non-dominant hand from males; (D) Spatial kernel density estimation plot generated by the FTT executed from non-dominant hand from males. (E) is the FTT frequency as a function of time for dominant hand from females; (F) Spatial kernel density estimation plot generated by the FTT executed from dominant hand from females; (G) is the FTT frequency as a function of time for non-dominant hand from females; (H) Spatial kernel density estimation plot generated by the FTT executed from non-dominant hand from females. Black crosses represent valid taps, red crosses represent non-valid taps and red ellipsis are the 95% confidence ellipsis fitted to the data.

## Discussion

Handedness and sex are recognized factors that influence the performance in the FTT using different instruments [2]. The novelty of the present investigation is a detailed description of the influence of handedness and sex on global, temporal, and spatial parameters obtained from three spatial designs of smartphone-based FTT in adults.

The common findings in the three protocols were: (i) total number of taps, minimum frequency, total displacement, and bandwidth Y were improved by handedness; and (ii) median frequency was significantly influenced by an interaction between sex and handedness in which males using dominant hand had better performance in the FTT. For Protocol I, the more common result was the presence of a significant main effect of sex and handedness on the variables.

The use of smartphones as instruments to perform FTT is a revolutionary advance in the method because it enables the development of different designs of finger-tapping tests and enables the addition of new parameters to the classical analysis. In the present investigation, we designed three protocols for smartphone-based FTT. Protocol I is the most similar protocol compared to classical approaches because the participant had to tap in a specific place to be as fast and accurate as possible. The results we observed for temporal parameters are qualitatively like those previously observed using different technologies: males had better performance than females and dominant hands had better performance than nondominant hands.

**Table 3. Descriptive statistics [median (interquartile range)] of the global, temporal and spatial parameters for Protocol III.**

| | Female | | Male | | |
|---|---|---|---|---|---|
| | **DH** | **NDH** | **DH** | **NDH** | **Significant effect size** |
| *Global parameters* | | | | | |
| Total number of taps | 99 (43.50) | 86 (31.25) | 111 (29.25) | 91.5 (19.75) | Handedness* |
| Number of errors | 1 (3.25) | 1.5 (8.75) | 2 (8.25) | 3 (9) | None |
| *Temporal parameters* | | | | | |
| Median frequency | 3.60 (1.04) | 3.10 (0.89) | 3.79 (1.08) | 3.25 (0.62) | Handedness*, Interaction** |
| Maximum frequency | 6.25 (1.12) | 4.98 (0.6) | 4.99 (1.13) | 4.83 (0.24) | None |
| Minimum frequency | 2.20 (0.83) | 2.06 (0.56) | 2.30 (1.01)[9] | 1.79 (0.42) | Handedness* |
| *Spatial parameters* | | | | | |
| Total displacement | 4489.62 (1884.95) | 4061.68 (1253.50) | 4999.15 (1508.71) | 4138.80 (876.3) | Handedness* |
| Major axis 1 | 4.92 (1.04) | 4.96 (0.87) | 4.74 (1.10) | 4.91 (1.01) | Handedness***, Interaction*** |
| Minor axis 1 | 3.64 (1.53) | 3.29 (1.25) | 2.63 (1.05) | 2.96 (0.84) | Sex* |
| Area 1 | 57.65 (32.00) | 50.33 (30.28) | 40.40 (20.50) | 46.08 (16.98) | Sex*** |
| Major axis 2 | 5.1 (0.86) | 4.92 (1.89) | 4.76 (1.38) | 5.8 (0.62) | None |
| Minor axis 2 | 3.51 (1.66) | 2.78 (1.34) | 2.57 (1.04) | 3.34 (0.91) | Sex**, Handedness* |
| Area 2 | 56.41 (33.32) | 41.58 (34.92) | 36.98 (22.72) | 59.58 (23.24) | Sex**, Handedness*** |
| Peak density area 1 | 6.01 (4.00) | 6.64 (4.90) | 6.09 (6.18) | 4.89 (2.98) | None |
| Bandwidth X area 1 | 61.92 (17.93) | 69.90 (28.62) | 62.80 (27.19) | 71.18 (19.18) | Handedness* |
| Bandwidth Y area 1 | 114.28 (45.38) | 123.76 (36.83) | 123.42 (39.37) | 141.46 (47.54) | Handedness* |
| Peak density area 2 | 6.55 (3.86) | 7.81 (5.27) | 6.64 (4.86) | 5.38 (3.61) | None |
| Bandwidth X area 2 | 61.96 (24.54) | 68.31 (28.23) | 63.90 (20.25) | 71.37 (23.30) | Handedness* |
| Bandwidth Y area 2 | 116.20 (40.70) | 127.67 (43.34) | 130.60 (46.71) | 139.32 (43.97) | Handedness* |

DH: dominant hand; NDH: nondominant hand.

*p < 0.001

**0.001 ≤ p < 0.01

***0.01 ≤ p < 0.05

The other two protocols we designed (protocols II and III) cannot be compared to the classical settings of FTT, because they are composed of tests that request alternated taps in two neighbor regions, and few investigations using smartphone-based FTT are comparable to these designs. The smartphone-based FTT can be grouped in tests that use one target [9] and two targets [9–16]. The tests that used two targets also differed from the task for tapping using one or two fingers for alternating movements, and only Lee et al. [4] have a more detailed description of the spatial dimensions of the test design.

All these studies using smartphone-based FTT evaluated the performance of patients with some diseases, especially Parkinson's disease, and did not emphasize the influence of handedness and sex on performance. The present investigation is a first approach to describe the basic features of the smartphone-based FTT in the adult population.

Dominant hands consistently are reported as having better performance than non-dominant hands in FTT [2]. Cortical correlates seem to be associated with the different performance considering hand dominance. Some findings indicate that left-handers are less lateralized than right handers [23,24]. The difference in the FTT performance between sexes was usually reported [25], but it is not clear why males and females differ in FTT performance.

The lack of left handers and aged persons are limitations of the present study and future study focusing on them might be interesting. Considering that use of touchscreen technology

is relatively new, more information can be extracted in future besides different designs to improve specific motor skills during finger tapping test.

Overall, the FTT is a useful tool for evaluating motor and cognitive function in a variety of diseases and conditions. Smartphone-based FTT has potential applications in the same diseases that were previously evaluated using other technologies such as Parkinson's disease, Alzheimer's disease, multiple sclerosis, traumatic brain injury, attention deficit hyperactivity disorder, and depression. Previous investigations had shown significant reliability of smartphone-based FTT [22] that encourage for more clinical applications, additional studies of validation comparing its performance with the gold-standard technologies such as video capture, and studies comparing the performances using different smartphone models (and touchscreen technologies).

Smartphone-based FTTs represent an opportunity to apply an important neuropsychological test in a wider range of the population, including the poorest people with less access to more sensitive medical instruments. Even considering that sex and handedness have partial influence on the smartphone-based FTT variables, our results reinforce the needs to create norms for males and females with additional consideration about the hand dominance, or create models to compensate the differences associated to these factors. More basic and clinical studies for FTT based on smartphones can help in assessing the variables that could optimize its diagnostic power.

## App availability

The a trial version of the app is available at the Play Store: https://play.google.com/store/apps/details?id=com.MomentumTouch

## Supporting information

**S1 Data. Raw files of all participants including the moment of the tap, x and y coordinates and the indication of the area where the tapping occurred.**
(ZIP)

## Author Contributions

**Conceptualization:** Felipe André Costa Brito, Anselmo de Athayde Costa e Silva, Givago Silva Souza.

**Data curation:** Givago Silva Souza.

**Formal analysis:** Felipe André Costa Brito, Luis Carlos Pereira Monteiro, Enzo Gabriel Rocha Santos, Givago Silva Souza.

**Funding acquisition:** Givago Silva Souza.

**Investigation:** Felipe André Costa Brito, Ramon Costa de Lima, Bruno Lopes Santos-Lobato, Givago Silva Souza.

**Methodology:** Felipe André Costa Brito, Bruno Lopes Santos-Lobato, André Santos Cabral, Bianca Callegari, Anselmo de Athayde Costa e Silva, Givago Silva Souza.

**Project administration:** Bianca Callegari, Anselmo de Athayde Costa e Silva, Givago Silva Souza.

**Software:** Felipe André Costa Brito, Enzo Gabriel Rocha Santos.

**Supervision:** Anselmo de Athayde Costa e Silva, Givago Silva Souza.

**Visualization:** Luis Carlos Pereira Monteiro, Givago Silva Souza.

**Writing – original draft:** Felipe André Costa Brito, Givago Silva Souza.

**Writing – review & editing:** Luis Carlos Pereira Monteiro, Enzo Gabriel Rocha Santos, Ramon Costa de Lima, Bruno Lopes Santos-Lobato, André Santos Cabral, Bianca Callegari, Anselmo de Athayde Costa e Silva, Givago Silva Souza.

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
