## [Decision Letter · Decision Letter 0]

22 Feb 2023

PDIG-D-23-00033

THE ROLE OF SEX AND HANDEDNESS IN THE PERFORMANCE OF THE SMARTPHONE-BASED FINGER-TAPPING TEST

PLOS Digital Health

Dear Dr. Givago Silva Souza,

Thank you for submitting your manuscript to PLOS Digital Health. After careful consideration, we feel that it has merit but does not fully meet PLOS Digital Health's publication criteria as it currently stands. Therefore, we invite you to submit a revised version of the manuscript that addresses the points raised during the review process.

Please submit your revised manuscript within 60 days Apr 23 2023 11:59PM. If you will need more time than this to complete your revisions, please reply to this message or contact the journal office at digitalhealth@plos.org. Please include the following items when submitting your revised manuscript:

We look forward to receiving your revised manuscript.

Kind regards,

Haleh Ayatollahi, PhD

Section Editor

PLOS Digital Health

Journal Requirements:

b. If any authors received a salary from any of your funders, please state which authors and which funders.

2. Figures 1, 2, 4, and 6 contains screenshots. We are not permitted to publish these under our CC-BY 4.0 license; websites are usually intellectual property and are copyrighted. This includes peripheral graphics of the web browser such as the buttons. We ask that you please remove or replace it.

Additional Editor Comments (if provided):

The topic of the manuscript is interesting. I appreciate if the authors address the following issues in their revision.

1- For keywords, please consider using appropriate MeSH terms.

2- In the introduction section, please add more information about the Finger-Tapping Test (FTT).

3- In the introduction section, please expand justification for conducting the research. 

4- In the methods section, please provide the ethics approval code.

5- In the methods section, please explain why only right-handed participants were included? What about the left-handed people?

6- How did the researchers recruit the participants? Was there any facilitators to guide the participants during completing their tasks? 

7- In the methods section, please separate the information for each protocol using appropriate subheadings.

8- Please follow the journal instructions for organizing the list of the references. Moreover, references older than 10 years ago need to be updated.

Reviewers' comments:

Reviewer's Responses to Questions

**Comments to the Author**

1. Does this manuscript meet PLOS Digital Health’s publication criteria? Is the manuscript technically sound, and do the data support the conclusions? The manuscript must describe methodologically and ethically rigorous research with conclusions that are appropriately drawn based on the data presented.

Reviewer #1: Yes

Reviewer #2: Yes

2. Has the statistical analysis been performed appropriately and rigorously?

Reviewer #1: Yes

Reviewer #2: Yes

3. Have the authors made all data underlying the findings in their manuscript fully available (please refer to the Data Availability Statement at the start of the manuscript PDF file)?

Reviewer #1: Yes

Reviewer #2: Yes

4. Is the manuscript presented in an intelligible fashion and written in standard English?

Reviewer #1: Yes

Reviewer #2: Yes

5. Review Comments to the Author

Reviewer #1: Overall, this is a clearly written and interesting manuscript that advances scientific knowledge in this field. I recommend publication but have some suggested edits to improve the manuscript.

General comments:

1. There are several instances of the incorrect abbreviation 'FFT' being used instead of 'FTT' - please correct this.

2. There is a lot of data presented in the Results section - the Forest plots are helpful but it is difficult to visualise the significance of the effects (particularly in the black and white figures sent). Perhaps statistical significance could also be incorporated into the tables - for example, highlighting in bold or using an asterisk system to indicate where the significant differences are?

3. The figure legends and descriptions in the text do not appear to match the figure labels in Figures 2, 4 and 6 (e.g. "Figure 2E shows an example of the spatial distribution of the taps for a representative participant" - it doesn't, it shows the female dominant hand tapping frequency)

Specific comments:

INTRODUCTION

4. Paragraph 2 - please correct grammar, e.g. "Initially, it was very common TO use THE telegraph key..." and, "... image-based motion [7], AND musical instrument midi keyboard [8]".

5. Paragraph 3 - please add further detail on the studies in references 15-22 so the reader does not have to search through the individual references. For example, what conditions were studied, in what contexts? More detail could also be added on the potential applications of the Finger-Tapping Test here and in the Discussion.

METHODS

6. Final paragraph - please correct grammar, e.g. "Figure 7 INDICATES that handedness..."

DISCUSSION

7. Paragraph 1 - 'Adult people' is an odd term - please change to 'adults'.

8. Paragraph 2 - define handedness here, i.e. use of the dominant hand (so this does not get confused with left- and right-handedness). This might also need to be clarified in other sections as I first interpreted handedness to mean left- vs. right-handedness.

9. Paragraph 2 - point (ii) would be clearer written as, "median frequency was significantly influenced by an interaction between sex and handedness..."

10. More detail could be added on the potential applications of the Finger-Tapping Test. Parkinson's Disease is mentioned frequently but are there other relevant conditions/clinical uses of the test? How does the FTT using a smartphone compare (in terms of accuracy and reliability) with FTTs using other technologies? Are there any limitations specific to smartphones? If there have not been studies on this, then this is an obvious area for future research.

Reviewer #2: PDIG-D-23-00033

Dear Editors, 

Att: THE ROLE OF SEX AND HANDEDNESS IN THE PERFORMANCE OF THE SMARTPHONE-BASED FINGER-TAPPING TEST. 

The proposed manuscript is a summative report of the role of sex and handedness in the performance of the smartphone–based FTT. In my opinion, the paper lacks clarity in writing due to specific reasons:

1. Upfront, the authors should state that an abbreviated version was used in the study. Please say why you chose an abbreviated version. What was the rationale for selecting the referenced abbreviated version? Do not let the reference do the work for you. Motivate your reasons for the tools/ instrument for this study. 

2. The significance should be indicated earlier in the paper. The rationale for the novelty or significance of this test needs to be stated in the article. 

3. Indicate the test’s administration time and include the breaks between the tests for each protocol. It is not evident from the design the duration of the test. 

4. Remember to indicate whether this was a patient self–report on handedness. 

The authors need to tend to the writing of the abstract and the critical measurements not foregrounded at the beginning of the paper or in the method section. It needs to be apparent from findings the clinical and statistical significance of the role of sex and handedness. Without justification or rationale for the measurement, the authors lose the reader on why sex and handedness differ in practice for this clinical group of patients as opposed to the theoretical assumption of the validity of measuring sex for the adapted version of the FTT. You cannot only state that it is significant for future reference. 

I propose that the authors address the lack of clarity in writing of the above-referenced feedback. Therefore, minor revision is required to consider for publication.

6. PLOS authors have the option to publish the peer review history of their article (what does this mean?). If published, this will include your full peer review and any attached files.

**Do you want your identity to be public for this peer review?** For information about this choice, including consent withdrawal, please see our Privacy Policy.

Reviewer #1: Yes: Sarah Buckingham

Reviewer #2: No

---

## [Decision Letter · Decision Letter 1]

6 Jun 2023

PDIG-D-23-00033R1

THE ROLE OF SEX AND HANDEDNESS IN THE PERFORMANCE OF THE SMARTPHONE-BASED FINGER-TAPPING TEST

PLOS Digital Health

Dear Dr. Souza,

Thank you for submitting your manuscript to PLOS Digital Health. After careful consideration, we feel that it has merit but does not fully meet PLOS Digital Health's publication criteria as it currently stands. Therefore, we invite you to submit a revised version of the manuscript that addresses the points raised during the review process.

Please submit your revised manuscript within 30 days Jul 06 2023 11:59PM. If you will need more time than this to complete your revisions, please reply to this message or contact the journal office at digitalhealth@plos.org. Please include the following items when submitting your revised manuscript:

We look forward to receiving your revised manuscript.

Kind regards,

Haleh Ayatollahi

Section Editor

PLOS Digital Health

Journal Requirements:

2. We have noticed that you have uploaded Supporting Information files, but you have not included a list of legends. Please add a full list of legends for your Supporting Information files after the references list. 

3. Please ensure that Funding Information and Financial Disclosure Statement are matched.

Additional Editor Comments (if provided):

Reviewers' comments:

Reviewer's Responses to Questions

**Comments to the Author**

1. If the authors have adequately addressed your comments raised in a previous round of review and you feel that this manuscript is now acceptable for publication, you may indicate that here to bypass the “Comments to the Author” section, enter your conflict of interest statement in the “Confidential to Editor” section, and submit your "Accept" recommendation.

Reviewer #1: (No Response)

Reviewer #2: All comments have been addressed

2. Does this manuscript meet PLOS Digital Health’s publication criteria? Is the manuscript technically sound, and do the data support the conclusions? The manuscript must describe methodologically and ethically rigorous research with conclusions that are appropriately drawn based on the data presented.

Reviewer #1: Yes

Reviewer #2: Yes

3. Has the statistical analysis been performed appropriately and rigorously?

Reviewer #1: Yes

Reviewer #2: Yes

4. Have the authors made all data underlying the findings in their manuscript fully available (please refer to the Data Availability Statement at the start of the manuscript PDF file)?

Reviewer #1: Yes

Reviewer #2: Yes

5. Is the manuscript presented in an intelligible fashion and written in standard English?

Reviewer #1: Yes

Reviewer #2: Yes

6. Review Comments to the Author

Reviewer #1: Thank you for making the recommended changes. This has greatly improved the paper but there are just a couple of minor corrections that are still needed:

• The incorrect abbreviation FFT is still used in the figure descriptions.

• Where there are significant effects stated in Table 3, it would be helpful to include p-values (either exact values or something like p < 0.01 or p < 0.001). This could be added to the table or as a footnote with significant values indicated by asterisks in the table.

There are still a few grammatical errors and I feel that the paper would benefit from some further proof-reading.

Reviewer #2: The writers addressed the comments and provided adequate feedback to the editor and reviewers.

7. PLOS authors have the option to publish the peer review history of their article (what does this mean?). If published, this will include your full peer review and any attached files.

**Do you want your identity to be public for this peer review?** For information about this choice, including consent withdrawal, please see our Privacy Policy. 

Reviewer #1: Yes: Sarah Buckingham

Reviewer #2: No

---

## [Decision Letter · Decision Letter 2]

20 Jun 2023

THE ROLE OF SEX AND HANDEDNESS IN THE PERFORMANCE OF THE SMARTPHONE-BASED FINGER-TAPPING TEST

PDIG-D-23-00033R2

Dear Mr Souza,

We are pleased to inform you that your manuscript 'THE ROLE OF SEX AND HANDEDNESS IN THE PERFORMANCE OF THE SMARTPHONE-BASED FINGER-TAPPING TEST' has been provisionally accepted for publication in PLOS Digital Health.

Best regards,

Haleh Ayatollahi

Section Editor

PLOS Digital Health

Reviewer Comments (if any, and for reference):

Reviewer's Responses to Questions

**Comments to the Author**

1. If the authors have adequately addressed your comments raised in a previous round of review and you feel that this manuscript is now acceptable for publication, you may indicate that here to bypass the “Comments to the Author” section, enter your conflict of interest statement in the “Confidential to Editor” section, and submit your "Accept" recommendation.

Reviewer #1: All comments have been addressed

2. Does this manuscript meet PLOS Digital Health’s publication criteria? Is the manuscript technically sound, and do the data support the conclusions? The manuscript must describe methodologically and ethically rigorous research with conclusions that are appropriately drawn based on the data presented.

Reviewer #1: Yes

3. Has the statistical analysis been performed appropriately and rigorously?

Reviewer #1: Yes

4. Have the authors made all data underlying the findings in their manuscript fully available (please refer to the Data Availability Statement at the start of the manuscript PDF file)?

Reviewer #1: Yes

5. Is the manuscript presented in an intelligible fashion and written in standard English?

Reviewer #1: Yes

6. Review Comments to the Author

Reviewer #1: (No Response)

7. PLOS authors have the option to publish the peer review history of their article (what does this mean?). If published, this will include your full peer review and any attached files.

**Do you want your identity to be public for this peer review?** For information about this choice, including consent withdrawal, please see our Privacy Policy.

Reviewer #1: **Yes: **Sarah Buckingham
